# Portal Vein and Mesenteric Artery Thrombosis Following the Administration of an Ad26.COV2-S Vaccine—First Case from Romania: A Case Report

**DOI:** 10.3390/vaccines10111950

**Published:** 2022-11-18

**Authors:** Florin Savulescu, Cristian Cirlan, Madalina Ionela Iordache-Petrescu, Mihai Iordache, Alexandra Bianca Petrescu, Cristian Blajut

**Affiliations:** 1Department of Medical-Surgical Specialities, “Titu Maiorescu” University of Bucharest, 040441 Bucharest, Romania; 2Central Military University Emergency Hospital “Dr. Carol Davila”, 010242 Bucharest, Romania

**Keywords:** portal vein thrombosis, mesenteric artery thrombosis, as26.COV2-S vaccine, feeding ileostomy, intestinal ischemia

## Abstract

COVID-19 has significantly affected public health, social life, and economies worldwide. The only effective way to combat the pandemic is through vaccines. Although the vaccines have been in use for some time, safety concerns have still been raised. The most typical adverse effects of receiving a COVID-19 vaccine are localized reactions near the injection site, followed by general physical symptoms such as headaches, fatigue, muscle pain, and fever. Additionally, some people may experience VITT (vaccine-induced immune thrombotic thrombocytopenia), a rare side effect after vaccination. We present the case of a 60-year-old female patient that developed VITT-like symptoms with spleno-portal thrombosis and intestinal ischemia two weeks after the administration of the Ad26.COV2-S vaccine. Surgical treatment consisted of extensive bowel resection with end jejunostomy and feeding ileostomy. Two weeks after the first operation, a duodenal-ileal anastomosis was performed. The patient was discharged five weeks after the onset of the symptoms. Although some rare adverse effects are associated with the SARS-CoV-2 vaccines, the risk of hospitalization from these harmful effects is lower than the risk of hospitalization from COVID-19. Therefore, recognizing VITT is significant for ensuring the early treatment of clots and proper follow-up.

## 1. Introduction

Since the COVID-19 pandemic began in 2019, it has changed the lives of people all over the world. Vaccines to prevent infection from its causative organism, SARS-CoV-2, were developed at the end of 2020 and are now our best hope for returning to normal by acquiring immunity against COVID-19 [1,2,3,4,5]. Because of the COVID-19 pandemic, drug companies have felt intense pressure to create effective and safe vaccines [6]. Because of this, clinical trials have been given a time limit to make them available faster and to help with the health crisis. However, as vaccines have been distributed globally at an unprecedented rate, there have been more and more reports of serious adverse events after vaccination. In particular, cases of vaccine-induced thrombotic and ischemic events such as immune thrombotic thrombocytopenia (VITT) have been recorded since February 2021, primarily after receiving the ChAdOx1 nCoV-19 vaccine (one of many adenovirus vector-based vaccines) [4,5,7,8,9,10,11,12,13,14,15,16,17]. VITT is caused by immunoglobulin G molecules that recognize platelet factor 4 (PF4) bound to platelets, which eventually causes platelet activation and the stimulation of the coagulation system; these antibodies are detectable through a PF4 enzyme-linked immunosorbent assay (ELISA) [8,18,19,20,21]. Although it is not caused by heparin exposure, the disease process is similar to heparin-induced thrombocytopenia (HIT) [5,22,23,24,25,26]. The clinical profile of VITT has not been completely elucidated either, but cases have involved various organ systems, such as cerebral veins, pulmonary arteries, portal veins, and peripheral veins. 

Some studies have reported a VITT association between vein thrombosis and arterial ischemia [27,28,29,30]. Venous and arterial thrombotic disorders have been traditionally considered separately because of their distinct anatomical differences and clinical presentations. Arterial thrombosis is typically caused by platelet activation, while venous thrombosis is mainly a result of clotting system activation.

Here, we describe the case of a patient with the portal vein and mesenteric thrombosis following the administration of a Ad26.COV2-S vaccine. To our knowledge, this is the first case published from Romania with this presentation. 

## 2. The Case Presentation

A 60-year-old previously healthy woman presented to the Emergency Department of the Central Military University Emergency Hospital “Dr. Carol Davila” in Bucharest, Romania, complaining of diffuse abdominal pain and a lack of intestinal peristalsis, which began a week prior. From her otherwise clean medical history, we noted that two weeks before the presentation, she was vaccinated with a Johnson & Johnson vaccine (Ad26.COV2-S). At clinical examination, she presented with a painful, distended abdomen with positive Blumberg’s sign and had no bowel movement for two days. A melena-type liquid was observed at the rectal examination. The lab tests were as follows: WBC: 22.05 k/microL, platelets (PLT): 86 k/microL, glucose: 348 mg/dL, CRP: 216.17 mg/L, D-dimer: 16,939 ng/mL, INR: 1.44, Quick time: 15.9, prothrombin activity: 60%, and activated partial thromboplastin time (aPTT): 55.8 s. The full lab tests are available in Table 1.

The patient’s CT scan revealed: an extensive thrombosis of the portal system, with the absence of contrast opacification of the portal vein—caliber of up to 22 mm, an absence of the opacification of intrahepatic portal branches, splenic vein and mesenteric veins, thrombosis of the splenic artery with extended spleen infarction, stasis liquid at the level of the esophagus that appears distended, a large amount of stasis liquid at the level of the stomach, jejunal loops starting from the angle of Treitz, swollen, with circumferentially thickened walls, overturned, bordered by an inflammatory infiltrate and liquid effusion, some necrotic–non-iodophilic, suggesting mesenteric ischemia, liver with perfusion disorders, collapsed inferior vena cava, and a pancreas with normal position, clear outline, and homogeneous structure. (Figure 1, Figure 2 and Figure 3).

At the time of the diagnosis, we did not have the CT-angiography-reconstructed images of the mesenteric artery available. A vascular surgeon was consulted, and after seeing the patient and the CT scan, he did not indicate SMA thrombectomy.

The patient had negative PCR COVID-19 test results. The patient’s past medical and family histories were unremarkable, and there was no personal or family history of thromboembolic events. No allergic reactions were reported in previous vaccinations. No known clinical risk factors for thrombosis or ischemia could be identified. Following hematological consultation, the patient was suspected of having a VITT-like syndrome, in the absence of thrombocytopenia. Unfortunately, the hospital could not test for anti-PF4 antibodies, so we cannot provide data on that front.

The diagnosis of proximal bowel ischemia was established, and the patient was rushed into the operating room. The procedure began 3 h after the patient’s initial presentation in the Emergency Department. The surgical exploration of the abdomen revealed a necrosed jejunum, starting at 5 cm distal to the angle of Treitz and continuing for 200 cm. The rest of the small bowel appeared to normal, but with diminished bowel movements. The colon appeared normal. The spleen had color modification but without any necrotic tissue in the splenic capsule (Figure 4).

We did not have available the indocyanine green (ICG) substance. So, the resection limit was decided based on the color of the intestine and the pulsations of the superior mesenteric artery branches. After resection of the ischemic small intestine, an end Folley jejunostomy and an end feeding ileostomy were performed. We considered performing a jejunoileal anastomosis, but because the blood supply of the remaining bowel was questionable, we decided not to perform the anastomosis. The end-feeding ileostomy also provided a visual inspection of the small bowel and an assessment of the progression or not of the ischemia. We did not perform a splenectomy since we did not know of the remaining functional splenic tissue. As for portal thrombosis, we planned on administering anticoagulation drugs, because unfortunately, endovascular treatment for portal thrombosis was unavailable at that time. After surgery and admission to the ICU, heparin treatment was started four hours after the procedure (a bolus of 5000 units and then a continuous infusion of 1300 units/h). Intravenous nutrition was started on Day 1. Enteral feeding was started on day two via the end ileostomy. A combination of Fresubin Intensive by Fresenius Kabi (70%) and bile from jejunostomy (30%) was used per patient tolerance. On day three, the patient regained bowel movements. In addition, the patient’s glucose level normalized on day three. On day 15, we performed a CT scan that showed minimal peripheral permeabilization of the portal trunk, but extensive venous thrombosis involving the portal vein and intrahepatic branches was maintained. The ileal loop at the ileostomy site was in good shape, with well-vascularized mucosa, so we decided to perform the anastomosis. After the procedure, drains were placed in the Douglas Pouch to guard for anastomotic leaks and bleeding. A naso-ileal catheter was placed to continue the enteral feeding; the catheter was removed after three days when the patient regained bowel movements. The postoperative course after the second operation was uneventful regarding the regaining of bowel movements, but it was complicated by the development of postoperative evisceration that was addressed by the debridement and re-closure of the abdomen. We did not use any mesh, as the risk of infection was very high. The patient was discharged 35 days after the onset of symptoms, with only coumarin anticoagulant as the medication. During the follow-up, two months after the onset of symptoms, the patient received a CT scan with oral and intravenous contrast that showed no oncological tumors. In addition, the patient received an upper-GI endoscopy and a colonoscopy that did not demonstrate tumors. The three-month follow-up showed an incisional hernia, and the patient was scheduled for surgery in the following weeks.

## 3. Discussion 

In the past few months, information about this new clinical entity dubbed VITT has come to light. This condition can be deadly, especially in young and healthy patients. VITT is a severe but rare complication that has occurred in some people after receiving the AstraZeneca or Johnson & Johnson COVID-19 vaccines [4,31,32]. These safety concerns developed because many individuals received these vaccinations during the pandemic. However, studies show no link between VITT and the BioNTech Pfizer vaccine [33,34,35]. Additionally, one case report exists of fatal thrombotic events after vaccination with Moderna [36]. It is unclear if the VITT was directly responsible for the patient’s condition or if it may have been related to a pre-existing illness [37]. A case report concerning 220 definite or probable victims of VITT in the UK stated that, on average, these patients were 14 days post-vaccination when diagnosed. This time frame ranged from 5 to 48 days. Furthermore, this study showed that female patients comprised 55% of all cases. Therefore, being female is a significant risk factor for VITT. VITT has been linked to an increased risk of arterial thrombosis, middle cerebral artery stroke, and occlusion of the peripheral arteries. The thrombosis sites of our patient include the superior mesenteric artery. The median platelet count for patients with VITT is 20,000–25,000 [38]. However, in this report, our patient’s platelet count was greater than 150,000. More than half of the patients in those studies had multiple thrombi in varying locations [39,40,41,42,43]. The patient in our case report shared similar characteristics with these other patients—she presented the symptoms 14 days after getting vaccinated, is female, and had multiple sites affected by thrombosis.

The case definition criteria for vaccine-induced immune thrombocytopenia and thrombosis, as decided in a study published by Pavord S. et al. [23], and also discussed in other studies [13,14,16,17], are found in Table 2. For a definite VITT diagnosis, a patient should meet all five of the following criteria: (1) the onset of symptoms 5–30 days after vaccination against SARS-CoV-2 (or less than 42 days in patients with isolated deep-vein thrombosis or pulmonary embolism), (2) the presence of thrombosis, (3) thrombocytopenia (PLT < 150,000 per cubic millimeter), (4) D-dimer levels of greater than 4000 FEU, and (5) positive anti-PF4 antibodies from ELISA. 

Our patient did not have thrombocytopenia at the presentation, and we did not have the ELISA anti-PF4 antibodies test available, so we established the diagnosis of Probable VITT (according to the Expert Hematology Panel [23]). In addition, at the initial diagnosis, we could not test for protein C and protein S deficiencies, or deficiencies in factor V, factor VII, or factor VIII. However, after we discharged the patient, these tests were performed, and the results showed no deficiencies.

In addition, due to partial thrombosis of the superior mesenteric artery causing bowel ischemia, we began the treatment of extensive portal vein thrombosis after the large ileal enterectomy. To manage VITT, treatments such as anticoagulation, high-dose intravenous immune globulin (IVIG), therapeutic plasma exchange (TPE), corticosteroids, rituximab, and eculizumab have been used [18,20,23,37,44,45,46,47,48]. In addition, many patients received transfusions of supportive blood products such as platelets and fibrinogen concentrate or cryoprecipitate. Because our patient did not meet the criteria for definitive VITT, the treatment of thrombosis included anticoagulation and corticosteroids, as per hospital protocols, without high-dose intravenous immune globulin (IVIG). 

Regarding the surgery, after the large enterectomy, we were faced with a dilemma. One could argue that we could have performed an anastomosis. However, we weigh the benefit of the anastomosis (faster return of the bowl to its normal physiology) with the possible complication (anastomotic fistula or peritonitis) and the impossibility of appreciating the ongoing ischemia of the small bowel. In this extreme situation, we decided to perform an end ileostomy (5 cm from the Treitz angle) with the help of a Foley catheter and an alimentary ileostomy. This allowed us to drain the stomach and the bile, feed the patient until the improvement of her thrombosis, and see the mucosa of the proximal ileum, which gave us information about the progression of the arterial thrombosis. A Witzel feeding ileostomy would have allowed us to feed the patient even for a couple of months until her nutritional status would have been optimal. However, with this procedure, we could not observe the vascularization of the ileal bowel. After the treatment of the patient’s thrombosis started and we saw its benefits, we decided to perform the anastomosis to return the bowl to its normal physiology. After improving the patient’s condition, we consider that the risk of anastomosis fistula should be less than at the previous surgery.

Patients with VITT and hepatosplenic thrombosis generally had lower platelet counts, higher D-dimer levels, and more thrombosis sites than patients without hepatosplenic thrombosis. Our patient, who had a normal platelet count at initial diagnosis and high D-dimer levels, does not meet this criteria. These findings suggest that hepatosplenic thrombosis is relatively more severe in VITT cases. More follow-up studies with larger sample sizes and more comprehensive clinical data are needed to improve the prognosis of hepatosplenic thrombosis in VITT through timely intervention.

Another issue that has to be discussed is the new diagnosis of diabetes mellitus [49,50,51]. At the initial presentation, the patient had a serum glucose level of 349 mg/dL. After the treatment began for her thrombosis, the glucose levels dropped to normal in 3 days. They stayed at that level without any treatment, allowing us to conclude that arterial thrombosis may also affect the pancreatic arteries, impairing insulin secretion and elevating the plasma glucose concentration, as has been found in other studies [52,53].

## 4. Conclusions

The available data from clinical trials and country surveillance programs show that COVID-19 vaccines effectively prevent serious illness, hospitalization, and death [36,54,55]. Long-term or severe side effects from these vaccines are infrequent. However, thrombotic and ischemic events have been reported after vaccination with adenoviral COVID-19 vector vaccines. Although they are sporadic, the course may be fulminant; therefore, clinicians should be familiarized with the VITT’s clinical and laboratory features and the recommended treatment methods [56,57,58,59]. Our patient presented with a case of probable VITT, which is a rare but serious complication that can arise from some COVID-19 vaccines. If not caught early and treated properly, VITT can be deadly. In this instance, the patient was successfully managed through surgery for ischemic events and medical intervention for the thrombotic events without any further issues related to low platelet counts, and was discharged fully recovered. VITT is an infrequent complication of the adenoviral-vector-based COVID-19 vaccines, and although it can result in serious medical issues, the benefits of protection against COVID-19 heavily outweigh any associated risks.

## Figures and Tables

**Figure 1 vaccines-10-01950-f001:**
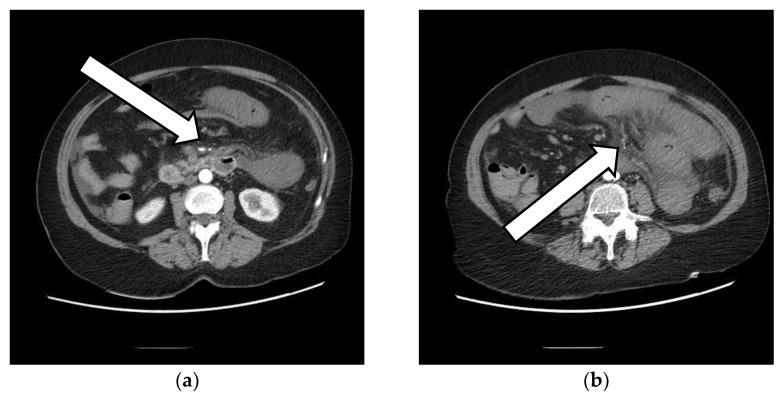
Arterial CT scan showing superior mesenteric artery. (**a**) Arrow is pointing toward the first branch of the superior mesenteric artery that appears to receive normal contrast; (**b**) Arrow is pointing toward distal branches of the superior mesenteric artery that tend to have thrombotic characteristics.

**Figure 2 vaccines-10-01950-f002:**
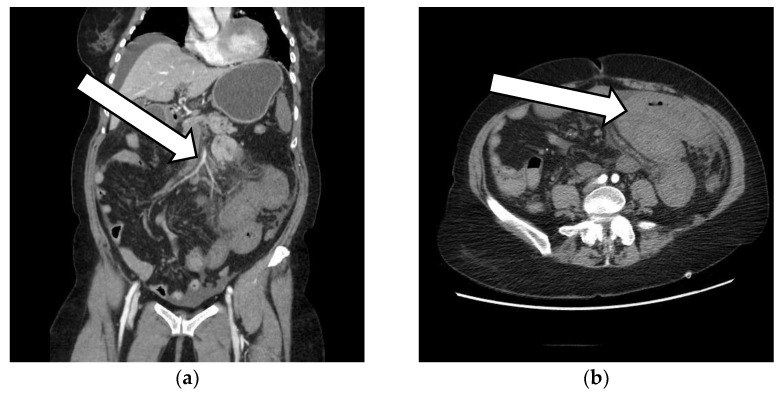
CT scan at admission. (**a**) Arrow is pointing toward the first branches of the superior mesenteric artery that appears to receive normal contrast, but with decreasing intensity; (**b**) Arrow is pointing toward the distended first loops of the jejunum, with thickened walls that do not appear to receive contrast.

**Figure 3 vaccines-10-01950-f003:**
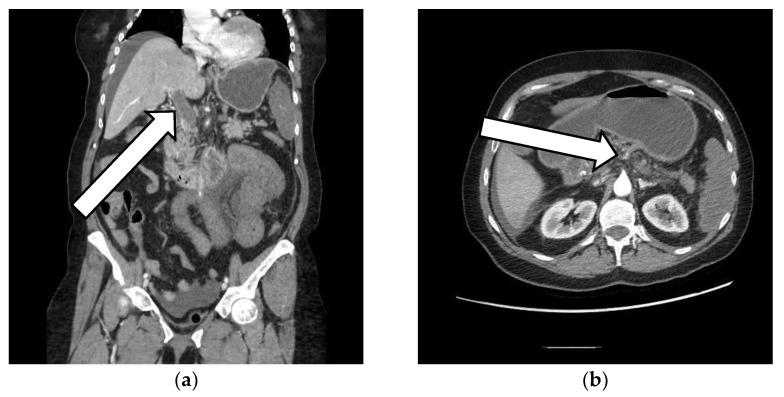
CT scan at admission. (**a**) Arrow is pointing toward the portal vein; one can appreciate the absence of contrast opacification of the portal vein and the absence of opacification of intrahepatic portal branches. (**b**) Arrow is pointing toward the splenic artery; one can appreciate the thrombosis of the splenic artery with extended spleen infarction.

**Figure 4 vaccines-10-01950-f004:**
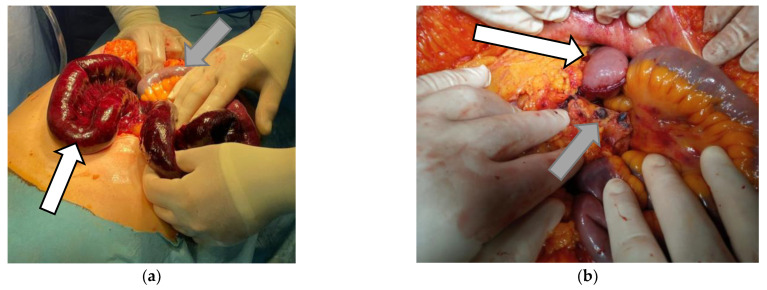
The intraoperative aspect of the bowels. (**a**) the white arrow is pointing toward the ischemic bowel; the grey arrow is pointing towards the normal-looking bowel. (**b**) The white arrow is pointing towards the jejunal stump; the grey arrow is pointing towards the venous thrombus inside the branches of the superior mesenteric vein.

**Table 1 vaccines-10-01950-t001:** Lab tests of the patient at admission. Only modified values have been recorded here.

Test	Value	Units
WBC	22.05	k/microL
Lymphocytes	6.7	%
Granulocyte	88.2	%
Eosinophils	0	%
Monocytes	1.07	k/microL
PLT	186	k/microL
Hemoglobin	12.3	g/dL
Serum glucose	349	mg/dL
Urea	45	mg/dL
Creatinine	1.33	mg/dL
Glutamyl transpeptidase	237	U/L
C reactive protein	216.17	mg/L
D-dimer	16,939	ng/mL

**Table 2 vaccines-10-01950-t002:** Case Definition Criteria for Vaccine-Induced Immune Thrombocytopenia and Thrombosis (VITT), According to an Expert Hematology Panel. Pavord S. et al. [23].

Type of VITT	Description
Definite VITT	All five of the following criteria:
	Onset of symptoms 5–30 days after vaccination against SARS-CoV-2 (or ≤42 days in patients with isolated deep-vein thrombosis or pulmonary embolism)
	Presence of thrombosis
	Thrombocytopenia (platelet count < 150,000 per cubic millimeter)
	D-dimer level > 4000 FEU
	Positive anti-PF4 antibodies from ELISA
Probable VITT	D-dimer level > 4000 FEU but one criterion not met (timing, thrombosis, thrombocytopenia, or anti-PF4 antibodies), D-dimer level unknown, or 2000–4000 FEU and all other criteria met
Possible VITT	D-dimer level unknown or 2000–4000 FEU with one other criterion not met, or two other criteria not met (timing, thrombosis, thrombocytopenia, or anti-PF4 antibodies)
Unlikely VITT	Platelet count < 150,000 per cubic millimeter without thrombosis with D-dimer level < 2000 FEU, or thrombosis with platelet count > 150,000 per cubic millimeter and D-dimer level < 2000 FEU, regardless of anti-PF4 antibody result, and alternative diagnosis more likely

## Data Availability

Not applicable.

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
