# Peer review of "Portal Vein and Mesenteric Artery Thrombosis Following the Administration of an Ad26.COV2-S Vaccine—First Case from Romania: A Case Report"

_vaccines, 2022, doi:10.3390/vaccines10111950_

Round 1

Reviewer 1 Report

Thank you for the opportunity to review this work. A few comments:

- Please provide a reference for the statement in the introduction "clinical trials have been given a time limit to make them available faster 38 and help with the health crisis"

- Why can't the hospital perform the PF4 test? Typically hospitals without a test can at least send out the specimen for testing at another facility. Was this not an option?

- Why was she taken to surgery emergently? The exam described does not suggest peritonitis. Was endoluminal thrombolysis an option to maybe restore flow? I understand it was not available for the portal vein, but what about for the SMA? Was vascular surgery consulted?

Author Response

Thank you for the review.

  1. We added "Accelerating vaccine trials. Bull World Health Organ. 2021 Jul 1;99(7):482-483. doi: 10.2471/BLT.21.020721. PMID: 34248219; PMCID: PMC8243025." as reference 59. Our statement does not want to imply literally that the drug companies were given a time limit but rather that, given the evolvement of the pandemia at that time, special measures were needed so that the vaccine could be available to the public as fast as possible.
  2. Our hospital can not perform the PF4 test. The hospital has a contract with a private lab for the tests that are unavailable in-house. That lab also can not perform an anti-PF4 antibodies test.
  3. The clinical presentation, with positive Blumberg's sign, and CT scan indicative of proximal mesenteric ischemia, allowed us to diagnose proximal jejunal infarction. As seen on the intraoperative images, the proximal bowel loops were unsalvageable. Endoluminal thrombolysis could have been possible, but only if the patient presented to the hospital earlier, before the necrosis of the bowels became irreversible. The CT scan did not show SMA thrombosis, and the flow of contrast substance was normal in the main branches of the SMA. So, only the small branches that supplied the proximal jejunum were thrombosed. We consulted the vascular surgeon, and after seeing the patient and the CT scan, he didn't indicate SMA thrombectomy.

The manuscript was modified so that it responds to your review.

Thank you. 

Reviewer 2 Report

Thrombocytopenia and thrombosis are serious side effects associated with COVID-19 vaccination, which have been extensively reported especially in people vaccinated with adenovirus vector vaccine. The pathophysiology of VITT is still not fully understood. This article provides a case diagnosed as probable VITT and the patient was finally given successful therapy, which has good reference value.

I suggest that if possible, more detailed clinical case data should be provided, such as whether there is thrombosis in other parts of the body, including the brain, lung, etc. Second, provide more imaging and laboratory results to rule out diseases that can cause arteriovenous thrombosis, such as oncological diseases, rheumatic immune diseases, hematological diseases, etc. 

Author Response

Thank you for the review.

  1. CT scan of the brain was not done. The patient did not have any neurological disorder at the time of the presentation at the hospital. CT scan of the lungs, abdomen, and pelvis did not show any other thrombosis apart from that noted in the manuscript. During the follow-up, two months after the onset of symptoms, the patient received a CT scan with oral and intravenous contrast that showed no oncological tumors. Also, the patient received an upper-GI endoscopy and a colonoscopy that did not demonstrate tumors. Also, a hematologist was consulted. After testing for protein C and protein S deficiency or deficiency in factor V, factor VII, or factor VIII, we haven't found any deficiency, as stated in the manuscript. 

We added the new clinical data to the manuscript. 

Thank you.

Reviewer 3 Report

Information is interesting, well presented and should be published. Paper needs a significant shortening; it is too long

Author Response

Thank you for the review.

We agree with you. The article may be too long for a "normal" case report. This article, discussing possible side effects of a Covid-19 vaccine, needs to emphasize certain aspects because we want to avoid being interpreted that we consider the vaccines unsafe. Apart from the case report, the manuscript also contains information about VITT, a disease that clearly needs more studies and care reports to be clearly defined and understood. The longest part of this article, the Discussion chapter, helps us show how our thinking process worked in this particular case. We reviewed the entire manuscript again and would like, with your consent, to keep all the information in it.

Thank you.

Round 2

Reviewer 3 Report

No attention to the reccomendation of shortening was paid